# Digital micro-banking as a health and protection intervention for street-connected children and youth? Analysis from a Togolese pilot

Neil Howard 

Social, University of Bath Faculty of Humanities and Social Sciences, Bath, UK

**Correspondence to**
Dr Neil Howard; nph28@bath.ac.uk

## ABSTRACT

**Background** The SaVa pilot offered street-connected children and youth (C&Y) access to digital micro-banking services to help them save money and thus avoid the violence associated with theft at night. It further used the micro-banking service as a gateway to attract C&Y towards other health and protection services. It took place over 12 months in Lomé, Togo and involved one social worker (SW) plus volunteers running 'the bank' in a non-governmental organisation-funded drop-in centre.

**Methods** C&Y peer researchers were trained to interview and use collective drawing with peers, which they did on a bi-monthly basis. The SW recorded information about case management, training and additional services offered. He administered a survey to C&Y at the point of entry into the project and after 1 year, asking questions about recent experiences of violence or theft. At the end of 12 months, the author conducted interviews with C&Y using the bank, C&Y researchers who used the bank, project staff and steering committee partners and child protection stakeholders. One focus group was conducted with C&Y participants, alongside limited participant observation of where C&Y sleep and work.

**Results** Findings suggest that the intervention was successful in encouraging children to open accounts and save money, and in contributing to a reduction in theft and associated violence. The incidence of theft reduced 90%; 95% of C&Y account holders were happy with the service; 200 accounts were opened. The project supported the delivery of additional services, including counselling, school reinsertion and removal from the streets.

**Conclusions** Analysis suggests that it is possible to offer street-connected C&Y formal financial services but that this offer has potential as a health and protection intervention, especially in combination with the provision of a safe space, dedicated SW support and add-on services.

## INTRODUCTION

There are millions of street-connected children and youth (C&Y) around the world and they are known to be vulnerable to violence, including theft and associated assault.[1] In 2017, the author led a participatory action research study with street-connected migrants and working C&Y in five West African cities, which found that theft, especially at night,

---

**WHAT IS ALREADY KNOWN ON THIS TOPIC**

⇒ There are millions of street-connected children and youth (C&Y) all around the world and they are known to be vulnerable to theft and associated violence. Lacking access to formal financial services, they typically rely on informal and often unreliable savings mechanisms which deepen their vulnerability.

**WHAT THIS STUDY ADDS**

⇒ This study shows that with the right institutional accompaniment, digital micro-banking is possible for street-connected C&Y and that it can contribute to increased savings and reduced theft and associated violence. It further shows that digital micro-banking can be used as an attractive 'hook' to bring C&Y into the orbit of additional services.

**HOW THIS STUDY MIGHT AFFECT RESEARCH, PRACTICE OR POLICY**

⇒ The findings strongly suggest that child protection actors should consider incorporating the offer of digital micro-banking services alongside more traditional offers of safe spaces, social worker support and service referral.

---

was among the greatest issues they faced.[2] Not only was theft traumatic and often accompanied by violence, but the fact that C&Y were so frequently robbed meant that they were effectively stuck in a poverty trap, unable to accumulate enough money to get off the streets or to go home. This dynamic is well-established in the literature, which shows that C&Y use multiple, often quite ingenious, informal avenues for saving—ranging from entrusting their money to a local adult to burying it overnight—but that none of these remain reliable in the long-term.[3–7] The literature further emphasises that there are substantial legal and cultural barriers to offering street-connected C&Y formal financial services. Lacking a fixed address or formal identification documents, few street-connected C&Y can meet the legal Know-Your-Customer requirements

underpinning most formal banking, including mobile money banking, while informal barriers such as discrimination on the part of bank staff or shame on the part of C&Y are also impactful.[3 8] Although there is some precedent for C&Y savings groups in Africa[9–13] and for pseudo-banking services for street-connected C&Y in South Asia,[4 14] the project's benchmarking study found no evidence either of the widespread provision of formal financial services to street-connected C&Y or of the use of digital services to this end.[3] Consequently, C&Y remain excluded from financial services and more likely to face theft, violence and the ongoing cycle of poverty.

The SaVa pilot emerged in this context and was implemented by the international child protection organisation, Terre des Hommes (Tdh). It sought to examine the theft-reduction and violence-reduction potential of providing these street-connected C&Y with access to digital micro-banking services in combination with more 'traditional' support such as a safe space, social worker (SW) accompaniment and referral. The hypothesis was that if C&Y had somewhere safe and accessible to put their money, they would be less likely to keep it on their person and thus less vulnerable to theft, while also being less likely to avail themselves of parallel informal savings mechanisms that often let them down. We also anticipated that children would be attracted to the technology of the micro-banking service, which would in turn act as a 'gateway' for additional child protection services. The research sought to establish the validity or otherwise of these expectations (Parts of this paper draw on and reproduce text from the project's evaluation[15]).

## METHODS

The SaVa model was straightforward. Operationally, it centred on the Tdh *Hope Point* at the Aflao border with Ghana. *Hope Points* are safe spaces for C&Y to come and rest, play or receive SW support (see figure 1 below). This one served the street-connected population-based around Lomé beach and was staffed by a senior SW. He was supported on a rotating basis by four experienced activists from the African Movement for Working Children and Youth (AMWCY), who did outreach work and contributed to the management of micro-banking services.

Day-to-day, the *Hope Point* operated like a bank, with C&Y coming to deposit and withdraw money during

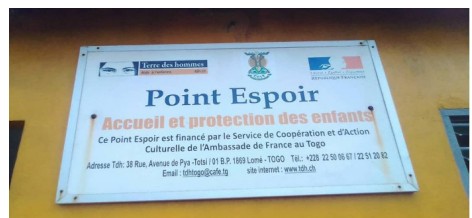

**Figure 1** The Hope Point drop-in centre. © Author—permission granted for use.

opening hours. Banking was provided by Ecobank in the form of an *Xpress Point*, which is an Ecobank-approved, subcontracted mobile money service. This bank included a counter, secured smartphones for transactions, and SIM cards for each individual account, which were kept in the *Hope Point* overnight to prevent loss or theft. Crucially, while Ecobank provided the financial infrastructure, Tdh acted as the legal sponsor for each account—in other words, each mobile money account and the SIM card associated with it was opened in Tdh's name (so as to enable the project and the bank to comply with Know-Your-Customer requirements), but Tdh kept a ledger in the *Hope Point* linking each SIM to their eventual C&Y 'owner'. This was a significant innovation. In addition, the SW, AMWCY activists and Ecobank staff offered basic financial literacy training to interested C&Y, while the *Hope Point* functioned as a place of refuge. Although it was already known to and used by C&Y, it was hoped that SaVa could attract more of them to it, with the bank serving as an entry point for orientation towards other necessary services such as healthcare or school reinsertion.

Research into the pilot sought to understand whether, how and for whom it 'worked', with which implications for child protection practice. Methods were mixed but lacked the representativity of a randomised control trial (RCT). C&Y researchers were trained to interview and use collective drawing with their peers, which they did on a bi-monthly basis. The SW recorded information on case management, training and additional services offered. He also administered a survey to all C&Y at the point of entry into the SaVa project and after 1 year, asked basic questions such as about recent experiences of violence or theft. At the end of the pilot's 12 months, the author conducted open-ended interviews with C&Y using the bank (6), C&Y researchers who used the bank (2), project staff and steering committee partners (10) and child protection stakeholders (4). One focus group was also conducted with C&Y participants (10), alongside limited participant observation of where C&Y sleep and work. This broad data set offers a solid base from which to interpret the project's findings. A full discussion of the project and its research methods can be found and freely accessed in the open-access evaluation.[15]

## RESULTS
### Financial matters

Consultations with Togolese child protection actors established a target of working with 150 C&Y over the year of the pilot. In terms of accounts opened, this was exceeded by over 30%, with 200 accounts opened and with the age range of account holders between 10 and 22 and the gender composition predominantly men (which reflects the wider demographics of the street-connected C&Y population in Lomé and across the region[2]). Total deposits surpassed 600 000 West African Franc (FCFA), close to 800 British Pounds (GBP), with withdrawals totalling around 440 000 FCFA. Almost 1400 transactions

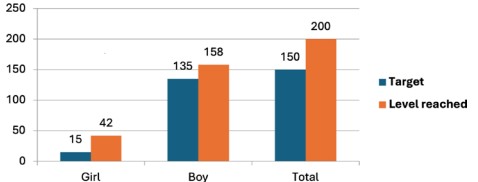

**Figure 2** Number of C&Y saving. Source: SaVa accounts data. C&Y, children and youth.

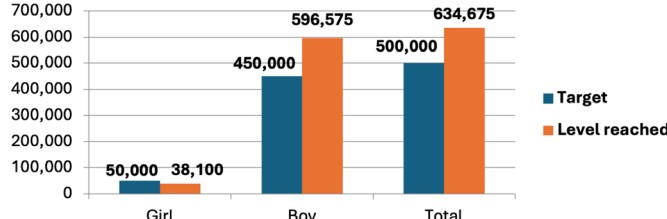

**Figure 4** Amount deposited. Source: SaVa accounts data.

took place, averaging around 400 FCFA per transaction, which is more than most C&Y spend on food in a day (see figures 2–5 below).

Although excitement was palpable at the existence of a bank 'just for them' and at the technology involved to run it (95% of users declared themselves 'happy' with it), C&Y did also report numerous difficulties in using it. Opening times—which were restricted to daylight hours—were often inconvenient, especially for C&Y busy in the day or making their money and wishing to deposit it at night. The lack of reliable internet access—on which the banking app depends—hampered various financial transactions. And the application itself, at the time relatively new, often crashed or experienced issues that prevented the processing of transactions.

These obstacles make the project's positive financial numbers appear even more impressive. Qualitative data suggest that they owe much to the efforts of the SW, the volunteers and the Ecobank staff, who conducted awareness-raising activities to promote the bank in the immediate vicinity of the *Hope Point* and further afield where street-connected, migrant and working C&Y were known to be concentrated. Football matches and music were also staged, while the promise of a (much-needed) free meal brought many in in the early days.

### A reduction in violence

C&Y using the bank reported a drastic reduction in theft and violence over the course of the project's pilot year. Fully 100% of the C&Y involved in the project reported experiencing some form of recent violence when opening their accounts at the start. After 6 months, this was down to 25% and at the end of the 12 pilot months, it was down below 10%—a 90% reduction, as can be seen in figure 6 below. Theft similarly plummeted. At the project's inception, the vast majority of participating C&Y reported having been robbed at some point over the preceding 6 months, often more than once. By contrast, at the end

of the pilot period, 67% of respondents had been 'theft-free' for the past 6 months.

Qualitative data echo these findings—in multiple interviews, participants confirmed that they no longer faced theft nor themselves stole. 'On ne vole plus rien maintenant' was a frequent refrain in this regard, and that phrase's double-meaning is important here—it can, and should, translate to 'we are not stolen from anymore' *and* 'we do not steal anymore'. The Tdh SW explained the cycle thus: when C&Y quickly spend their money for fear of losing it later, they at times end up with nothing and then find themselves resorting to theft in order to be able to eat. Evidently, the ability to access reliable savings mechanisms interrupted this cycle.

### Skills development, hope and safety

The opening of individual savings accounts presented the SW and *Hope Point* volunteers with an opportunity to offer C&Y trainings in financial literacy, numeracy and budgeting. These were complimented by guidance on how to develop 'a life plan'. Data suggest that this orientation was positively experienced by C&Y, many of whom reported the project giving them skills, 'a sense of hope' and 'something to work for'. Importantly, the psychological function of saving appears to have mattered for many, in that it gave C&Y an anchor in the future, something safe to work towards or through and thus a vehicle for their aspiration, which many translated into plans to accumulate enough savings to enter an apprenticeship. For others, as interview transcripts repeatedly made clear, it was the *Hope Point* itself that was the most important thing that the project offered, since it represented a place of refuge in lives that are hard, insecure and often vulnerable:

'Here is somewhere we can come and rest, sleep, be safe…' Interview with a 15-year-old boy.

'Here we can play, eat, hang out with our friends…' 15-year-old man, focus group participant.

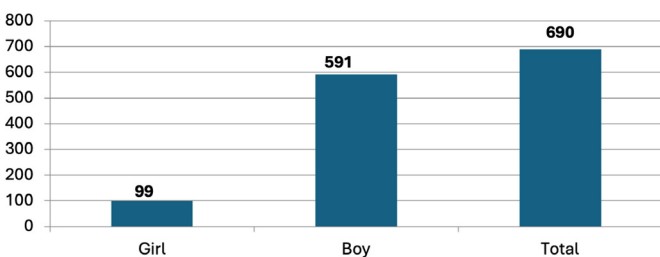

**Figure 3** Number of deposits. Source: SaVa accounts data.

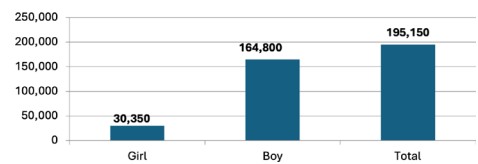

**Figure 5** Total savings. Source: SaVa accounts data.

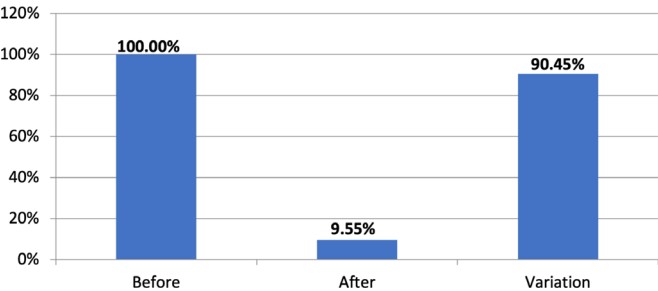

**Figure 6** Reductions in violence amongst SaVa participants. Source: SaVa social worker survey.

### A gateway for other services

As anticipated, the offer of digital micro-banking services to these C&Y acted as an entry point for providing other support and services. 54 C&Y, for example, were oriented towards social or health services according to their needs. Many benefitted from interpersonal or family mediation. Some received empathy when they came to the *Hope Point* with their difficulties; others advice when they needed it. As table 1 below makes clear, fully 111 C&Y went through situation-changing interventions during the life of the pilot, including 33 who left the streets.

### DISCUSSION

Evaluation of the SaVa pilot demonstrates, first and foremost, that it is possible to provide financial inclusion for street-connected C&Y—a population almost universally excluded from formal financial services. The SaVa model of partnering a bank and a child protection institution willing to act as the 'sponsor' complying with Know-Your-Customer legislation and then offering over-the-counter services appears unique. In this respect, it represents a potentially significant departure for C&Y financial inclusion, in particular for the highly vulnerable, typically marginalised group that is the street-connected.

Second, the reductions in violence and theft associated with the pilot are undoubtedly impressive, but it is impossible to attribute causality entirely to the innovation of micro-banking alone. Indeed, it is scientifically possible that these results are just chance, while there is an evident risk of respondent bias, with C&Y answering questions with what they think those asking would like to hear. Yet qualitative research squarely point in the direction of alternative, more promising explanations. First, those depositing their money in the digital micro-bank appear to have become less attractive to potential thieves and claimed as much themselves. Second, having a savings account reduced the imperative for C&Y to quickly spend what money they have and thus end up resorting to theft themselves. Third, and most importantly, data suggest that the sensitisation and diligent on-the-ground work by the SW and his volunteers were paramount in shifting social norms among C&Y away from theft and violence. For although C&Y would previously be robbed by gangs of youths or predatory adults, they would often simply steal from each other, since such behaviour was normalised. In this respect, SaVa's digital micro-banking represented a kind of 'hook' around which more traditional, SW-led behaviour change work could take place.

Our third essential finding is that while micro-banking is attractive and impactful for C&Y, part of its power lies in bringing them into the protective 'net' and thus opening a pathway for the delivery of important, parallel services. Building on this, we may argue that while the evaluation methodology does not enable full disaggregation of the relative impacts on C&Y of SW support, the provision of a safe space and the provision of banking services, this does not matter, since we can be highly confident that this *combination* is effective. Indeed, perhaps the major conclusion of our analysis must be that this combination 'works' for the vulnerable population in question. C&Y enjoyed having access to financial services and opening a savings account evidently supported the lengthening of their financial time horizons, which in turn had consequences for their ability to plan and aspire as well as to avoid quick spending and thus future theft. C&Y further enjoyed the safety and succour available at the *Hope Point* and the support provided by their SW, which included orientation to parallel services. In this respect, the study confirms that 'traditional' offers such as safe spaces and orientation to other services must continue to play a role in C&Y protection, but that digital micro-banking is a worthy addition to the protective mix. It further suggests that the effectiveness of digital micro-banking

| Table 1 Secondary services provided | | | |
|---|---|---|---|
| **C&Y supported through hope point** | | | |
| **Target group** | **Boys** | **Girls** | **Total** |
| Children put back in school | 19 | 0 | 19 |
| Children kept in school | 08 | 07 | 15 |
| Children taken out of work and put in an apprenticeship | 13 | 06 | 19 |
| Children supported to stay in an apprenticeship | 02 | 06 | 08 |
| Children taken off the streets and reunited with their families | 33 | 17 | 50 |
| **Totals** | **75** | **36** | **111** |
| .C&Y, children and youth; Source, SaVa social worker data. | | | |

will depend on its being embedded in a traditional C&Y protection offering.

This study is evidently only a single case and a pilot at that; as such, its findings are necessarily limited. This is compounded by the fact that evaluation was not through an RCT, meaning that generalisability is impossible. Nevertheless, the findings are intriguing enough to suggest that digital micro-banking is possible for street-connected C&Y and that it can contribute to increased savings and reduced theft and associated violence. They further show that digital micro-banking can be used as an attractive 'hook' to bring C&Y into the orbit of additional services. The findings suggest that child protection actors should consider incorporating this offer alongside more traditional offers of safe spaces, SW support and service referral. The international child protection community would do well to replicate the SaVa model and conduct further research towards refinement.

## Key findings

There are millions of street-connected C&Y all around the world and they are known to be vulnerable to theft and associated violence. Lacking access to formal financial services, they typically rely on informal and often unreliable savings mechanisms which deepen their vulnerability. This study shows that with the right institutional accompaniment, digital micro-banking is possible for street-connected C&Y and that it can contribute to increased savings and reduced theft and associated violence. It further shows that digital micro-banking can be used as an attractive 'hook' to bring C&Y into the orbit of additional services. The findings strongly suggest that child protection actors should consider incorporating the offer of digital micro-banking services alongside more traditional offers of safe spaces, SW support and service referral.

**Acknowledgements** The author would like to acknowledge the contributions of the entire SaVa project team to the design and evaluation of the project.

**Contributors** NH is the sole author of this article.

**Funding** This work was supported by an Innovation Fund grant (Safe Savings – SaVa') from the international child protection organisation, Terre des Hommes Lausanne (Tdh): Avenue de Montchoisi 15, CH-1006, Lausanne, Switzerland. It partnered Tdh, Ecobank, the African Movement of Working Children and Youth (AMWCY) and the University of Bath (UoB).

**Competing interests** None declared.

**Patient and public involvement** Patients and/or the public were involved in the design, or conduct, or reporting, or dissemination plans of this research. Refer to the Methods section for further details.

**Patient consent for publication** Not applicable.

**Ethics approval** This project was granted ethical approval after an internal review by Tdh's project ethics advisor on 1 December 2018. Participants gave informed consent to participate in the study before taking part.

**Provenance and peer review** Not commissioned; externally peer reviewed.

**Data availability statement** Data are available upon reasonable request. Data may be obtained from a third party and are not publicly available. The original report on which this article was based is publicly available and free to download, cited in the submission. Certain of the primary data and the research tools can be shared once anonymised by the author. Others are owned by a third party and contain personal identifiers and cannot be shared for legal reasons.

**ORCID iD**
Neil Howard http://orcid.org/0000-0003-2328-8230

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
