## [Reviewer comments · BMJ Paediatrics Open]

ARTICLE DETAILS

TITLE (PROVISIONAL)	Digital Micro-Banking as a Health and Protection Intervention for Street-Connected Children and Youth? Analysis from a Togolese Pilot(For consideration in the Special Collection on Health and Wellbeing of Street and Working Children, in collaboration with the International Society for Social Pediatrics and Child Health (ISSOP))
AUTHORS	Howard, Neil

VERSION 1 – REVIEW

REVIEWER	Dr. Shona Macleod University of Nottingham, Rights Lab
REVIEW RETURNED	15-Jan-2024

GENERAL COMMENTS	This is an interesting article setting out the findings from an innovative project providing street-connected children and youth with access to microfinance, as a gateway to additional services. There are just a few areas where changes could further strengthen the paper: 1. In the Methods section of the Abstract, it is not clear to unfamiliar readers what 'the bank' refers to. Perhaps an additional line of context is required.2. In Key Findings (p.5 line 11)- should C&W be C&Y?3. In the introduction, consider spelling out briefly what the findings are.4. Consider engaging with existing literature on street-connected children's access to financial services/barriers to access. A brief background section would help readers who do not work in the street-connected children sector better understand the challenges the project sought to address.5. On p.6, what are "legal KYC requirements"?6. Why do C&Y choose to use this service instead of anywhere else that people deposit cash to mobile money accounts? While one can guess at various reasons, it would be helpful to mention explicitly why a specific service for street-connected C&Y is required.7. Are the gender differences in the results due to much higher numbers of boys on the street generally, or are there other factors?8. On p.9 – "which is important because when they burn through cash for fear of loss, they end up with nothing" – this needs more explanation or rephrasing9. On p. 9 – "data strongly suggest that the sensitisation and diligent on-the-ground work by the SW and volunteers were paramount in shifting social norms away from an acceptance of theft and violence." – it is not entirely clear whose acceptance of theft and violence is referred to here.
--

REVIEWER	Michael Goodman
REVIEW RETURNED	16-Jan-2024

GENERAL COMMENTS	Your study tackles a critical issue: reducing violence among street-connected children and youth through a novel micro-finance approach. This holds strong potential for impact. You demonstrate the feasibility of your intervention, which is exciting. Building on this, consider deepening your engagement with existing research on microfinance and savings, particularly studies like Ssewamala et al.'s work in Uganda. This will strengthen the theoretical framework and showcase your research within a broader context. To enhance the flow, consider shifting authorial interpretations and explanations to the Discussion section. This will allow readers to fully grasp the results before drawing conclusions with you. Similarly, integrating qualitative and quantitative data analysis in the Results section would showcase how they mutually inform each other and enrich the understanding of your research question. Providing clear details about participant numbers and characteristics (age range, demographics) will offer better context for your findings. Specifying the format of interviews (closed-ended or open-ended) will give readers additional insight into your data collection. The significant reduction in violence observed is promising, but it's also important to acknowledge it might seem unexpected to readers. In the Discussion, consider exploring alternative explanations beyond the microfinance intervention, such as potential social support networks or external factors. This fosters open discussion and strengthens the validity of your conclusions. Clearly defining key terms and measures in the Methods section will ensure clarity and understanding for readers. Addressing any potential concerns about mobile technology theft associated with the mobile money service would be helpful. Karimli, L., Ssewamala, F. M., Neilands, T. B., & McKay, M. M. (2015). Matched child savings accounts in low-resource communities: who saves?. Global Social Welfare, 2, 53-64. Ssewamala, F. M., & Ismayilova, L. (2009). Integrating children's savings accounts in the care and support of orphaned adolescents in rural Uganda. Social Service Review, 83(3), 453-472. Karimli, L., & Ssewamala, F. M. (2015). Do savings mediate changes in adolescents' future orientation and health-related outcomes? Findings from randomized experiment in Uganda. Journal of Adolescent Health, 57(4), 425-432. Lombe, M., & Ssewamala, F. M. (2007). The role of informal social networks in micro-savings mobilization. J. Soc. & Soc. Welfare, 34, 37.
--

VERSION 1 – AUTHOR RESPONSE

Reviewer: 1
 Dr. Shona Macleod, University of Nottingham

Comments to the Author

This is an interesting article setting out the findings from an innovative project providing street-connected children and youth with access to microfinance, as a gateway to additional services.

There are just a few areas where changes could further strengthen the paper:

1. In the Methods section of the Abstract, it is not clear to unfamiliar readers what 'the bank' refers to. Perhaps an additional line of context is required.

NOTED. DONE.

2. In Key Findings (p.5 line 11)- should C&W be C&Y?

YES, THANKS. CHANGED.

3. In the introduction, consider spelling out briefly what the findings are.

DONE.

4. Consider engaging with existing literature on street-connected children's access to financial services/barriers to access. A brief background section would help readers who do not work in the street-connected children sector better understand the challenges the project sought to address.

EXTRA LITERATURE HAS BEEN INCLUDED, WITH GREATER MENTION OF THESE BARRIERS EARLY IN THE PAPER.

5. On p.6, what are "legal KYC requirements"?

SPELLED OUT CLEARLY NOW, THANKS.

6. Why do C&Y choose to use this service instead of anywhere else that people deposit cash to mobile money accounts? While one can guess at various reasons, it would be helpful to mention explicitly why a specific service for street-connected C&Y is required.

DETAIL ADDED IN INTRODUCTION.

7. Are the gender differences in the results due to much higher numbers of boys on the street generally, or are there other factors?

A NITE HAS BEEN ADDED SPECIFYING THE GENDER COMPOSITION DIFFERENCE IN THE SAMPLE POPULATION (WHICH IS OVERWHELMINGLY MALE).

8. On p.9 – "which is important because when they burn through cash for fear of loss, they end up with nothing" – this needs more explanation or rephrasing

THANKS. RE-PHRASED.

9. On p. 9 – "data strongly suggest that the sensitisation and diligent on-the-ground work by the SW and volunteers were paramount in shifting social norms away from an acceptance of theft and violence." – it is not entirely clear whose acceptance of theft and violence is referred to here.

NOTED. POINT RE-PHRASED.

Reviewer: 2
Michael Goodman

Comments to the Author

Your study tackles a critical issue: reducing violence among street-connected children and youth through a novel micro-finance approach. This holds strong potential for impact.

You demonstrate the feasibility of your intervention, which is exciting. Building on this, consider deepening your engagement with existing research on microfinance and savings, particularly studies like Ssewamala et al.'s work in Uganda. This will strengthen the theoretical framework and showcase your research within a broader context.

THANK YOU. THIS HAS NOW BEEN DONE. THE READING RECOMMENDATIONS WERE VERY WELCOME AND HAVE BEEN CITED.

To enhance the flow, consider shifting authorial interpretations and explanations to the Discussion section. This will allow readers to fully grasp the results before drawing conclusions with you.

ALL DONE. THE PAPER HAS BEEN SUBSTANTIALLY RE-ORDERED TO INCLUDE ALL INTERPRETATION AND EXPLANATION IN THE DISCUSSION.

Similarly, integrating qualitative and quantitative data analysis in the Results section would showcase how they mutually inform each other and enrich the understanding of your research question.

MORE MATERIAL ADDED TO INTEGRATE AS SUGGESTED.

Providing clear details about participant numbers and characteristics (age range, demographics) will offer better context for your findings.

ADDED.

Specifying the format of interviews (closed-ended or open-ended) will give readers additional insight into your data collection.

ALSO ADDED, IN METHODS.

The significant reduction in violence observed is promising, but it's also important to acknowledge it might seem unexpected to readers.

In the Discussion, consider exploring alternative explanations beyond the microfinance intervention, such as potential social support networks or external factors. This fosters open discussion and strengthens the validity of your conclusions.

THANK YOU. I HAVE ATTEMPTED TO MAKE CLEARER HOW UNCERTAIN THE CONCLUSIONS ARE AND WHAT OTHER FACTORS MAY BE AT PLAY.

Clearly defining key terms and measures in the Methods section will ensure clarity and understanding for readers.

Addressing any potential concerns about mobile technology theft associated with the mobile money service would be helpful.

THANK YOU. ADDED IN THE METHODS SECTION.